# Frying Performance of Gallic Acid and/or Methyl Gallate Accompanied by Phosphatidylcholine

**DOI:** 10.3390/foods12193560

**Published:** 2023-09-25

**Authors:** Ghazaleh Sadeghi Vahid, Reza Farhoosh

**Affiliations:** Department of Food Science and Technology, Faculty of Agriculture, Ferdowsi University of Mashhad, Mashhad P.O. Box 91775-1163, Iran; gh.sadeghi230@gmail.com

**Keywords:** frying, gallic acid, methyl gallate, phosphatidylcholine, reverse micelles

## Abstract

This study shows the possibility of using gallic acid (GA) and/or methyl gallate (MG) accompanied by phosphatidylcholine (PC) instead of *tert*-butylhydoquinone (TBHQ) for frying purposes. The antioxidants and PC were added in the concentrations of 1.2 mM and 500–2000 mg/kg, respectively. Oxidative stability index (OSI) and the kinetics of change in conjugated dienes (LCD), carbonyls (LCO), and acid value (AV) were used to assess the antioxidative treatments. GA alone and GA/MG (50:50) plus PC at 2000 mg/kg yielded the same OSI as that of TBHQ (18.4 h). The latter was of the highest frying performance in preventing the formation of LCD (*r*_n_ = 0.0517/h and *t*_T_ = 10.6 h vs. *r*_n_ = 0.0976/h and *t*_T_ = 4.5 h for TBHQ), LCO (*r*_n_ = 0.0411/h and *t*_T_ = 12.7 h vs. *r*_n_ = 0.15/h and *t*_T_ = 4.3 h for TBHQ), and hydrolytic products (AV_m_ = 37.8 vs. 24.0 for TBHQ); *r*_n_: normalized the maximum rate of LCD/LCO accumulation; *t*_T_: the time at which the rate of LCD/LCO accumulation is maximized; AV_m_: quantitative measure of hydrolytic stability.

## 1. Introduction

Frying is one of the most well-known methods of food preparation in the food industry, due to its low price, high speed, and creating desirable taste, color, and crispiness. It is currently estimated to have an economic value of USD 82 billion for the process in the USA, and twice as much in the rest of the world [1]. Deep-fat frying provides a relatively harsh condition of a simultaneous transfer of mass and heat between the hot oil (>170 °C) and the foodstuff being fried, during which desirable and/or undesirable flavor compounds might be produced. Moreover, some changes are likely to occur in the quality of oil and its oxidative stability through certain deteriorative reactions, e.g., thermal oxidation, hydrolysis, and polymerization [2].

Antioxidants have frequently been used to stabilize frying oils. Synthetic and natural antioxidants are considered as the main influencing agents to scavenge free radicals that result from oxidative reactions. The side effects of synthetic antioxidants, such as propyl gallate (PG), butylatedhydroxyanisol (BHA), butylatedhydroxytoluene (BHT), and *tert*-butylhydoquinone (TBHQ), have been examined in different studies and their use has been questioned due to the corresponding toxicity and carcinogenicity [3]. Concerns related to the safety of food additives as well as the increase in the level of awareness of consumers have created a tendency to replace synthetic antioxidants with natural alternatives. In addition, natural antioxidants have frequently been shown to possess health-giving effects on humans [4].

Gallic acid (GA) and methyl gallate (MG) are among the natural phenolic antioxidants well-known for many of their valuable biological activities [5]. There are limited studies implying their activity at high temperatures. For example, the thermal oxidation of corn and soybean oils inhibited by TBHQ, BHA, PG, and GA for 8 h at 180 °C suggested GA as a potential natural substitute for the first three synthetic antioxidants to protect vegetable oils at high temperatures [6]. The oxidation of sunflower oil heated at 120 °C indicated higher efficacy of GA, MG, and their combinations than TBHQ for the inhibition of polar compound formation [7]. In a very recent study into frying at 180 °C, GA/MG combinations were shown to exert performances the same as TBHQ or even better than that [8].

Phospholipids are a class of amphiphilic lipids consisting of a polar phosphate headgroup attached to a hydrophobic backbone containing glycerol and fatty acids. There are many studies indicating the antioxidative and/or synergistic effects of phospholipids in bulk oils. Chelation of prooxidative metals by the phosphate headgroup [9], the production of antioxidative compounds through the Maillard reactions [10], the regeneration of primary antioxidants [11], and an increase in the accessibility of antioxidants to initiating radicals in reaction [12,13], are all often suggested to explain the supporting role of phospholipids in lipid oxidation. In general, phosphatidylcholine (PC) is the predominant phospholipid in vegetable oils, although some other types, such as phosphatidylethanolamine (PE), phosphatidylserine (PS), and phosphatidylinositol (PI), are usually found in phospholipid fractions [14]. Very few studies have been carried out on applying PC for frying purposes. For example, in a small-scale preliminary frying test at 185 °C, PC at 500 mg/kg offered no protection to the canola oil triacylglycerols in lowering the amount of degradation products; however, the phospholipid at higher concentrations (1000 and 2000 mg/kg) could significantly stabilize the oil system [15]. It was also observed that PC at concentrations above 500 mg/kg synergistically promoted the frying performance of dihydrocaffeic acid amide [16].

Despite much research focusing on lipid oxidation inhibited by the natural antioxidants GA and/or MG as well as PC at mild temperatures, the literature shows some limited studies separately on their potency under thermal or frying conditions. Hence, this study aimed to evaluate the performance of GA and/or MG accompanied by PC in an institutional deep-fat frying condition. To do this, the kinetics of chemical changes during frying at 180 °C protected by the antioxidants in the presence of PC (500, 1000, and 2000 mg/kg) and TBHQ (a synthetic antioxidant of powerful antioxidant activity) were studied.

## 2. Materials and Methods

### 2.1. Materials

Refined, bleached, and deodorized oils (sunflower and palm olein) without any antioxidant additives were supplied by Segol factory, Nishabour, Iran. The vegetable oils were stored at −18 °C until analysis. The potatoes of Agria variety were directly purchased from the farms in Fariman, Iran. Analytical-grade GA, MG, and PC (L-α-phosphatidylcholine) were purchased from Sigma-Aldrich (St. Louis, MO, USA). All other chemicals of analytical reagent grade were purchased from Merck (Darmstadt, Germany) and Sigma-Aldrich (St. Louis, MO, USA).

### 2.2. Initial Quality Indicators of the Vegetable Oils

An oil (0.3 g) solution in hexane (7 mL) was shaken with 2 N methanolic KOH (7 mL) at 50–55 °C for 15 min to form fatty acid methyl esters. After 5 min, anhydrous Na_2_SO_4_ was added to the upper layer and then filtered. The solution was analyzed by FAD-GC (Hewlett-Packard, Santa Clarita, CA, USA). A 60-m BPX 70 capillary column (internal diameter: 0.2 mm; film thickness: 0.2 mm) and helium gas (0.7 mL/min) were used for the separation. Oven, injector, and detector were heated to 198 °C, 280 °C, and 250 °C, respectively. The calculated oxidizability (COX) values were calculated by Equation (1) [17]:COX value = [C18:1 + 10.3(C18:2) + 21.6(C18:3)]/100(1)

Peroxide (PV, meq/kg) and acid (AV, mg of KOH to neutralize acids in one g of oil) values were determined according to the thiocyanate [18] and AOCS [19] methods, respectively. Total tocopherols (mg/kg, α-tocopherol as standard) and phenolics (mg/kg, GA as standard) contents were measured according to Wong et al. [20] and Capannesi et al. [21].

### 2.3. Oxidative Stability Index (OSI)

The OSIs (h) were determined by a Metrohm Rancimat (743, Herisau, Switzerland) for 3 g of the control oil (sunflower/palm olein 65:35) containing PC (500, 1000, and 2000 mg/kg), TBHQ (1.2 mM), or 1.2 mM of GA, MG, or GA/MG 50:50 plus PC at 500, 1000, or 2000 mg/kg at 110 °C and an airflow rate of 15 L/h.

### 2.4. Frying Procedure

The potatoes (7.0 × 0.5 × 0.3 cm) were peeled and cut and then submerged in water (25 °C). Prior to frying, the potato stripes were rinsed with cold water and dried by a fan and a clean towel as well. The potatoes (20 g) were fried (180 °C) in 1 L of the oil treatments at 7 min intervals for 8 h in a bench-top Hamilton fryer (230, Hubei, China). No replenishment was made. At 1 h intervals, the oils (10 g) were filtered and stored in the dark at 4 °C until analysis. Frying was carried out in triplicate. The total contents of lipid-peroxidation conjugated dienes (LCD, mM) [22] and carbonyls (LCO, μmol/g) [23] as well as AV [19] were monitored over frying.

### 2.5. Kinetic Data Analysis

Kinetic curves of LCD formation were plotted over time *t* (h). The sigmoidal Equation (2) was fitted on the kinetic LCD data [24,25]:(2)$$\mathrm{LCD}=\frac{a}{b+e^{a\left(c-t\right)}}$$
where *a*, *b*, and *c* are the equation parameters. LCD_max_ was the ratio *a*/*b*. At the equation’s turning point, *t*_T_ = (*ac* − ln*b*)/*a*, the rate of LCD formation (mM/h) reached *r*_max_ = 0.25*a*^2^/*b*. Normalized *r*_max_ (*r*_max_/LCD_max_ = *r*_n_, /h) was calculated from 0.25*a*. The parameters *a* and *b* represent the pseudo-first order composite rate constant *k*_c_ (/h) for the LCD formation and the pseudo-second order rate constant *k*_d_ (/mM h) for the LCD lost through chemical decomposition and/or the Diels–Alder reaction (Figure 1), respectively.

Kinetic curves of LCO formation were plotted over time *t*. The sigmoidal Equation (3) was fitted on the kinetic LCO data [26]:(3)$$\mathrm{LCO}=a+\frac{b}{1+e^{\left(\frac{c-t}{d}\right)}}$$
where *a*, *b*, *c*, and *d* are the equation parameters. LCO_max_ was *a* + *b*. At the equation’s turning point, *t*_T_ = *c*, the rate of LCO formation (μmol/g h) reached *r*_max_ = 0.25*b*/*d*. The normalized *r*_max_ (*r*_max_/LCO_max_ = *r*_n_, /h) was calculated from the ratio *b*/4*d*(*a* + *b*).

Kinetic curves of AV were plotted over time *t*. The power Equation (4) was fitted on the kinetic AV data:(4)$$\mathrm{AV}=a+bt^{c}$$
where *a*, *b*, and *c* are the equation parameters. The ratio 1/*bc* (AV_m_), representing quantitatively the change pattern of organic acids over time, was employed as an empirical measure of hydrolytic stability.

### 2.6. Statistical Analysis

All determinations, in triplicate, were subjected to analysis of variance (ANOVA). ANOVA and regression analyses were based on MStatC version 4.00 (Michigan State University, East Lansing, MI, USA) and SlideWrite version 7.0 (Advanced Graphics Software, Carlsbad, CA, USA). Significant differences were determined according to Duncan’s multiple range tests. *p* values below 0.05 were considered statistically significant.

## 3. Results

### 3.1. Characterization of the Vegetable Oils

The vegetable oils were of the fatty acid compositions in agreement with those reported in the literature [27,28] (Table 1). The palm olein oil was much more saturated (mainly C16:0 and C18:0) than the sunflower oil by a factor of about four. Instead, the unsaturation content, especially linoleic acid (C18:2), in the palm olein oil was remarkably lower than that in the sunflower oil. This naturally makes the latter a rather unstable system with respect to the oxidation order for C18:3, C18:2, C18:1, and C18:0 as 2500:1200:100:1 [29]. The quite bigger COX value of the sunflower oil compared with the palm olein oil (6.7 vs. 1.7) also clearly shows the higher susceptibility of the former to lipid peroxidation. Accordingly, a blend of the sunflower (65% *w*/*w*) and palm olein (35% *w*/*w*) oils, which indicates a higher frying stability [30], was applied to assess the frying performance of the frying systems.

The oils had PV < 2 meq/kg and AV < 0.3 mg/g, indicating their suitable initial quality (Table 1). The contents of tocopherols and phenolics as biologically active and antioxidant compounds were in the normal ranges for the corresponding commercial oils [31].

### 3.2. OSI Values

The OSI value of the control oil plus the antioxidative treatments is shown in Figure 2. This measure basically indicates the time resistance of a lipid system to the drastically increased accumulation of a number of volatile organic acids as secondary/tertiary oxidation products in harsh oxidative circumstances [32].

TBHQ, as expected, considerably increased the OSI of the control oil from 11.4 h to 18.4 h. PC alone showed no significant effect on the OSI of the control oil at 1000 mg/kg but acted as prooxidant and antioxidant at 500 mg/kg and 2000 mg/kg, respectively. As for the GA and/or MG treatments, the OSI value significantly increased as the PC concentration increased. GA plus PC at 2000 mg/kg gave an OSI value comparable to that of TBHQ. Such a result was obtained for GA alone in our previous research [8], indicating no synergistic effect of the combination regarding the formation of the oxidative volatile acids. However, PC generally improved the antioxidant activity of MG, which was of no increasing effect on the value of OSI in the previous study. This implies that PC may give rise to a better exploitation of relatively weaker antioxidants in protecting lipid systems.

### 3.3. Kinetics of Change in the Total LCD Content

LCD include a wide variety of oxidation products due to the shifts in the position and geometry of the double bonds in polyunsaturated fatty acids [33]. The measure, which is simple, quick, and fairly inexpensive to determine, correlates well with the total polar compounds as the most common analytical quantity to investigate the health of used frying oils [34]. Its content increases initially and attains a plateau at advanced levels of deterioration over frying. This is basically due to a balance between the rates of LCD formation and the loss of conjugation resulting from secondary oxidations [35] as well as the Diels–Alder dimerization reaction (Figure 1) [36]. Such a trend is in line with the sigmoidal function of LOOH accumulation developed recently [24,25].

The kinetic data from Equation (2) fitted (R^2^ > 0.97) on the LCD changes over the frying at 180 °C (Figure 3A) are summarized in Table 2. The level of LCD_max_, which is a function of the ratio between the rates of LCD formation (*k*_c_) and LCD lost (*k*_d_) by chemical decomposition and/or the Diels–Alder reaction [8,25] of the control oil (27.3 mM), was significantly changed by the antioxidative treatments. Naturally, more oxidatively stable oils provide LCD molecules of lower tendency to transformation in any way [24]. As can be seen in Table 2, only the GA/MG/PC2000 and GA/PC2000 treatments could significantly increase LCD_max_ to 35.5 mM and 30.1 mM, respectively. Considering the kinetic parameter *r*_n_, which gives us an oxidative measure to compare the frying performance of antioxidative compounds overall [25], the GA/MG/PC2000 treatment, followed by the GA/PC2000 or MG/PC2000 treatments, performed the best and even better than TBHQ. This was a fairly better frying performance than had already been observed for the GA and/or MG treatments alone [8]. In addition, the parameter *t*_T_ as a time threshold to sensory rejection of frying oils [37] revealed the antioxidative treatment performance in the order of GA/MG/PC2000 > GA/PC2000 > MG/PC2000 > TBHQ.

### 3.4. Kinetics of Change in the Total LCO Content

LCO include a range of volatile and non-volatile products that dramatically affect the sensory and nutritional qualities of edible fats and oils. A recent study on the kinetics of change in the total LCO content provided some valuable parameters for the evaluation of oxidative stability [26]. Carbonyls, in practice, show a sigmoidal pattern over the course of oxidative deteriorations, i.e., an initial slow-increasing stage and then a second rapid-rising stage terminated to a maximum level [26]. In the end, they remain constant or decrease essentially due to further degradations to the tertiary products of non-carbonyl character and/or of higher volatility than the parent carbonyls [38,39].

The kinetic parameters resulting from Equation (3) fitted (R^2^ > 0.97) on the changes in the total LCO content over the frying at 180 °C (Figure 3B) are summarized in Table 3. The kinetics of change in carbonyls provided us with more interesting information about the performance of the antioxidative treatments. The LCO_max_ of the control oil was remarkably decreased in the presence of TBHQ. This was consistent with the significantly reduced values of *r*_max_ and *r*_n_. Significantly similar change patterns were observed when adding GA or MG alone, plus the increased concentration of PC, although they performed better than TBHQ on the whole; however, the GA/MG/PC treatments yielded the increased values of LCO_max_ as the PC concentration increased. With respect to the decreased *r*_max_ values with an increase in the PC concentration, it could be concluded that primary carbonyls had undergone lesser amounts of degradations to non-carbonyls and/or more volatile products [38,39]. Given the values of *r*_n_ and *t*_T_, the highest performance against the growing production of carbonyl compounds over the process belonged to the GA/MG/PC treatments and then GA/PC~MG/PC treatments at the same PC concentrations.

To more comprehensively characterize the frying performance of the antioxidative treatments, the ratio between the sensory threshold parameters *t*_T_ (Table 2) and the total LCO contents at the time, called the overall carbonyls-based peroxidation indicator, OPI_C_, was calculated (Figure 4). This provides a more valuable quantitative measure to evaluate frying oils in the presence or absence of antioxidants/prooxidants with respect to the primary and secondary oxidations at the same time. As shown in Figure 4, TBHQ could significantly increase the value of the OPI_C_ in the control oil from 0.29 h g/µmol to 0.40 h g/µmol. In general, the GA and/or MG treatments containing PC at 500 or 1000 mg/kg appeared prooxidative or slightly antioxidative effects on the control oil. However, all the antioxidative treatments plus PC at 2000 mg/kg were able to remarkably improve the frying performance of the control oil in the order of GA/MG/PC2000 > GA/PC2000 > MG/PC2000 > TBHQ.

### 3.5. Kinetics of Change in AV

AV is one of the most common measures to detect the hydrolysis of triglycerides during frying. Frying oils of a higher AV are more prone to generate the off-flavors and toxicity due to the further degradations of the free fatty acids and glycerol [40,41].

The power Equation (4) fitted well (R^2^ > 0.97) the changes in the AVs over the frying at 180 °C (Figure 3C). Figure 5 exhibits the empirical measures AV_m_. TBHQ efficiently prevented hydrolysis of the frying oil and, therefore, the following nutritional and toxicological deteriorations [40,41]. PC significantly improved the anti-hydrolytic effect of GA, MG, and their combinations as its concentration increased, and the highest efficiency belonged to the GA/MG/PC treatments. It is noteworthy to mention that the GA/MG 50:50 treatment with no significant anti-hydrolytic effect during frying in our previous study [8] considerably improved the hydrolytic stability of the control oil when adding the increased concentrations of PC.

## 4. Discussion

The major phospholipid PC in vegetable oils [14] carries a quaternary amine choline, which is not likely to serve as a Millard reaction substrate [15]. This is sort of a privilege for PC over the other raised phospholipids, such as PE and PS with primary amine groups, which react with the carbonyls from the *β*-scission reactions [42] to generate brown color Millard reaction products, restricting them to some extent for frying purposes. However, the phospholipids with primary amine groups have been found to form some Millard reaction products with inhibitory effects on lipid oxidation [43,44].

Apart from the negative effect on color, another concern with applying phospholipids in frying oils is their contribution to foaming, according to Blumenthal’s surfactant theory of frying [45]. Nevertheless, no significant foaming was observed throughout the entire frying process when lecithin [46] and PC [16] were incorporated at concentrations lower than 2000 mg/kg. However, phospholipids have been shown to accelerate lipid oxidation by reducing the surface tension of the system, which, in turn, increases the rate of oxygen diffusion/miscibility into the oxidizing lipid [47]. The prooxidative activity of 500 mg/kg PC alone in the Rancimat test (the OSI values in Figure 2) and the treatments that lowered the OPI_C_ value (Figure 4) might be explained in this manner. Similar Rancimat results were obtained when adding lecithin at 1000 mg/kg to rapeseed oil [48], PC at 100–400 mg/kg to olive oil [49], and PC at ~215 mg/kg to corn oil [50].

The metal-binding capacity through the negatively charged phosphate group could actually provide further explanation on the prooxidative/antioxidative activity of phospholipids. Prooxidative phospholipids chelate transition metals of different reactivity and thereby increase their solubility in the lipid environment [51]. This may promote metal–lipid interaction and accelerate lipid oxidation via the chelated metals that have still retained their reactivity [52]. In general, phospholipids at concentrations exceeding that of a reactive metal are likely to exert multiple bindings, which tie up all the metal coordination sites and make the metal unreactive [53]. The increased protective effects of PC at higher concentrations can be attributed to the multiple binding of transition metals that are naturally present in vegetable oils. Moreover, they may be due to the decomposition of lipid hydroperoxides into relatively innocuous hydroxides and esters rather than into reactive aldehydes [54]. According to Ishikawa et al. [55], trimethylamine oxide resulting from the molecular cleavage of PC significantly reduced methyl linoleate peroxidation by decomposing lipid hydroperoxides to methyl keto-octadecadienoate.

Over the past decade, the interfacial phenomena have provided us with additional information regarding lipid oxidation and any involved extrinsic and/or intrinsic factors. On this basis, the amphiphilic phospholipids are capable of self-assembling into a range of association colloids in the bulk oils containing partial amounts of water. As the amount of water is raised in the oils rich in phospholipids, the association colloids can change from spherical reverse micelles to spherical swollen micelles, cylindrical or rodlike micelles, hexagonal aggregates, and then lamellar structures. In the bulk oils with about 200–800 ppm water, phospholipids normally form reverse micelles at quantities exceeding their critical micelle concentrations (CMC) [56]. Depending upon the water content, type of bulk oil experimented, presence or absence of minor components, and heterogeneities in acyl chain lengths and in the number, position, and configuration of double bonds, the CMC of PC has been determined to be in the range of 50–950 mg/kg [11,12,13,56,57,58]. Zhao et al. [12] suggested a threshold CMC value for PC (~500 and 1000 mg/kg in the bulk and stripped peanut oils, respectively) to act as an antioxidant/prooxidant, which was markedly higher than its analytically determined CMC (95 mg/kg). Given the normal contents of phospholipids in commercial oils (38–600 µM from 3300 µM in crude oils) [57] and also the amounts applied in this study (500–2000 mg/kg), the formation of reverse micelles in the oil treatments with an increase in the PC concentration should not be far from our expectation.

Reverse micelles create oil–water interfaces where many oxidative reactants, e.g., prooxidative metals, amphiphilic LOOH, and lipophilic triacylglycerols, are brought into close contact with each other, leading to increased rates of lipid oxidation [52]. There are some studies indicating that the reverse micelles formed by PC in stripped vegetable oils accelerated lipid oxidation [11,12,56,57,59]. However, bulk oils are heterogeneous systems containing indigenous primary antioxidants (especially tocopherols) that have been shown to be synergistic with phospholipids in inhibiting lipid oxidation, due basically to (1) forming antioxidative Maillard products with the help of tocopherols, (2) regenerating tocopherols, and/or (3) changing the physical positioning of tocopherols [52]. Considering the lack of primary or secondary amine groups, PC has not been recognized to be synergistic with tocopherols in the formation of antioxidative Maillard reaction products [49] as well as an antioxidant regenerator [11,60,61]. In contrast, several studies demonstrated that the reverse micelles formed by PC in bulk oils synergistically improved the antioxidant activity of tocopherols by bringing them into close proximity to the oil–water interface where iron-dependent oxidation reactions take place [12,14,62,63,64,65]. Therefore, the improved performances caused by PC at higher concentrations in the present study must have been due to the shifts in the physical location of the indigenous and also added antioxidants, making them close to the actual site of oxidation reactions. Looking generally into the different experimental results, PC at 2000 mg/kg was of the remarkably highest frying performance, which can be attributed to the significant formation of reverse micelles as the PC concentration is increased to quantities higher than 1000 mg/kg.

The significant formation of reverse micelles at 2000 mg/kg PC could also provide a logical explanation for the better antioxidant activity of GA than MG overall. Structurally, GA is more hydrophilic than MG (log P for GA = −0.21 and for MG = −0.14) [13], which accordingly makes it concentrated more at the oil–water interface where lipid oxidation prevalently occurs. Mansouri et al. [13] showed that the mixed PC reverse micelles formed at the end of the induction period in the stripped sunflower oil had a bigger particle size as well as led to higher values of interfacial tension in the presence of GA than MG, indicating a larger diffusion of the former into the reverse micelles. Cui et al. [11] reported that PC reverse micelles generated in the stripped soybean oil decreased the oxidizability of the lipid system to a greater extent in the presence of polar Trolox than in the presence of its nonpolar analog α-tocopherol of lower partitioning at the oil–water interface. Most importantly, PC reverse micelles could explain how the two antioxidants behaved synergistically. One of the potential ways to achieve synergism is for one antioxidant to relocate into the site of lipid oxidation and the second antioxidant to regenerate the primary antioxidant that is preferentially oxidized in the system. The primary antioxidant regeneration requires a higher potency of hydrogen transfer for the secondary antioxidant [66]. The more electron-donating methyl ester group in MG has been reported to make its phenolic hydroxyl group of a lower bond dissociation enthalpy (O–H BDE) than that of GA (70.9 vs. 71.5 kcal/mol). More potent antioxidants possess lower values of the thermodynamic parameter O–H BDE, facilitating the process of direct hydrogen transfer to a radical [67]. The synergism of this kind has already been reported for α-tocopherol and ascorbic acid as the primary and secondary antioxidants, respectively, during the oxidation of methyl linoleate [68].

## 5. Conclusions

This study was the first attempt to kinetically evaluate the protective effect of the natural polyphenols GA and MG accompanied by the major phospholipid PC in vegetable oils for frying purposes. As the concentration of PC increased, the frying performance of the antioxidant treatments improved potentially due to the following: (1) the multiple binding of prooxidative metals, (2) the decomposition of LOOH into relatively unreactive molecules, and (3) the shifts in the physical location of the antioxidants. The best frying performances were observed at 2000 mg/kg PC, when the formation of reverse micelles was of greater possibility. The possible existence of PC reverse micelles in turn provided us with a convincing explanation about the better antioxidant activity of GA than MG as well as the synergy between them. On the whole, the antioxidative treatments at 2000 mg/kg PC exerted frying performances significantly better than TBHQ. This undoubtedly enables the edible oil industry to produce more healthy frying oils by avoiding the worrying application of TBHQ, although some additional studies regarding various frying oils that are used to fry other food materials would be helpful for a better evaluation of the efficacy of the antioxidative treatments studied here.

## Figures and Tables

**Figure 1 foods-12-03560-f001:**
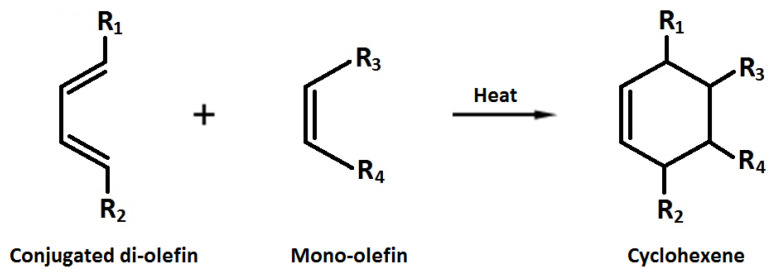
The Diels–Alder dimerization reaction.

**Figure 2 foods-12-03560-f002:**
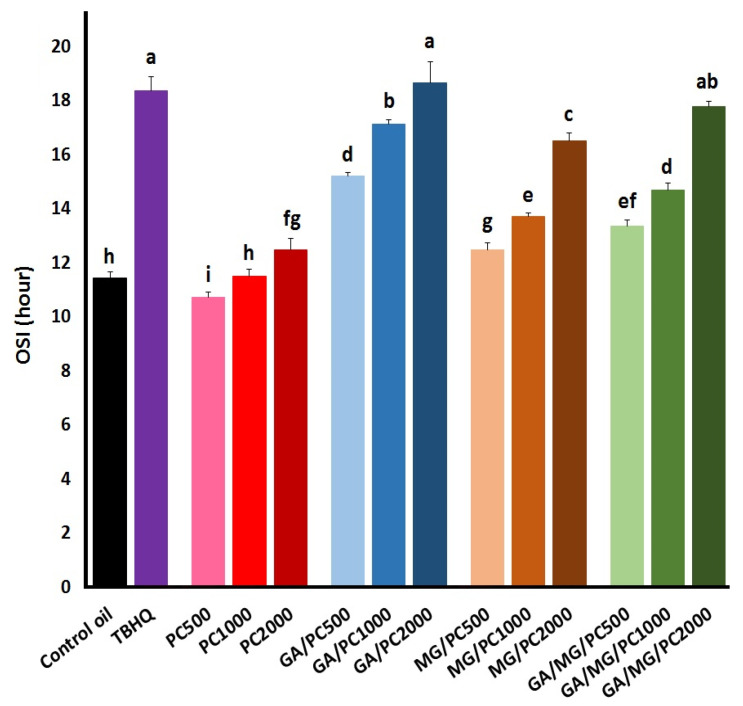
Oxidative stability index (OSI, h) of the control sunflower/palm olein (65:35) oil containing TBHQ (1.2 mM), PC (500–2000 mg/kg), or GA and/or MG (1.2 mM) plus PC. Means ± SD with the same lowercase letters are not significantly different at *p* < 0.05.

**Figure 3 foods-12-03560-f003:**
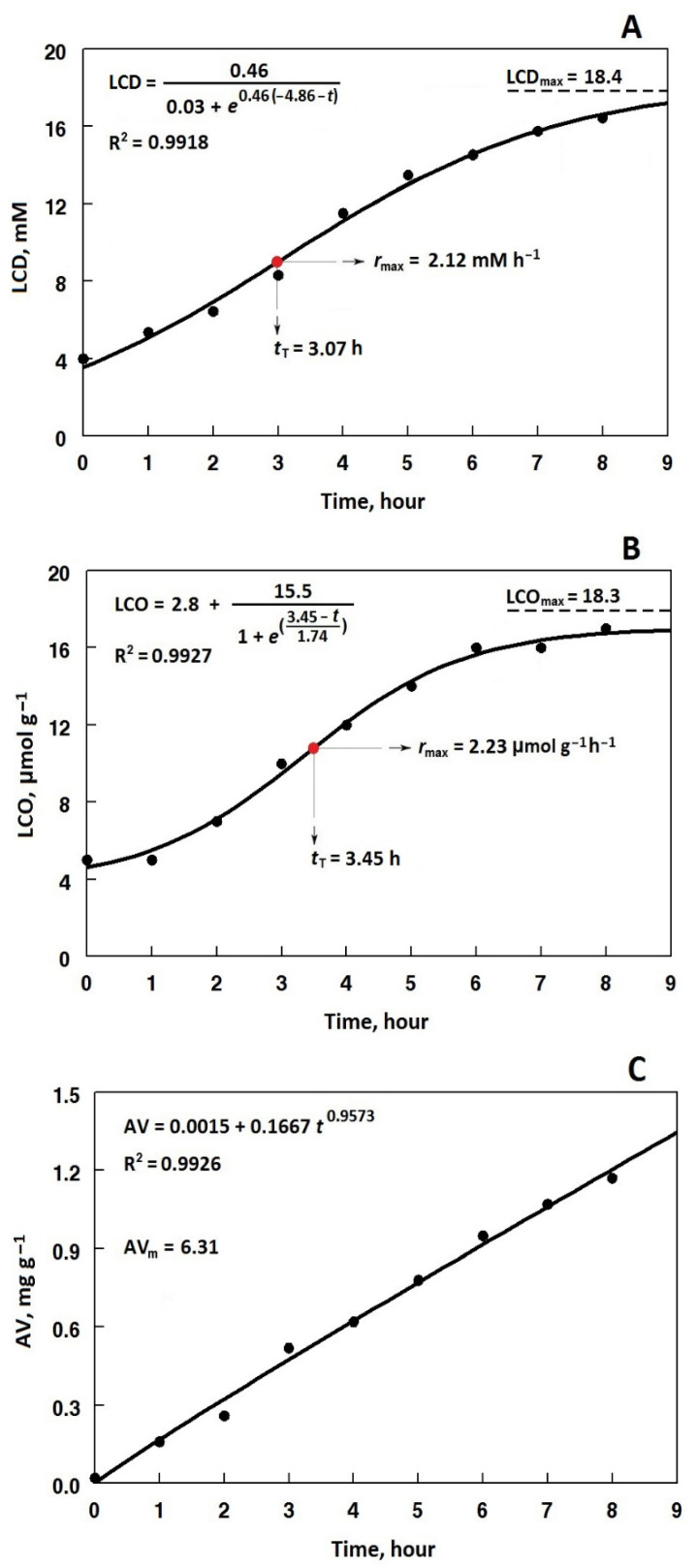
Kinetic curves of the corresponding LCD (**A**), LCO (**B**), and AV (**C**) during frying of the control sunflower/palm olein (65:35) oil containing GA (1.2 mM) plus PC (1000 mg/kg) at 180 °C, and the kinetic parameters from Equations (2) to (4).

**Figure 4 foods-12-03560-f004:**
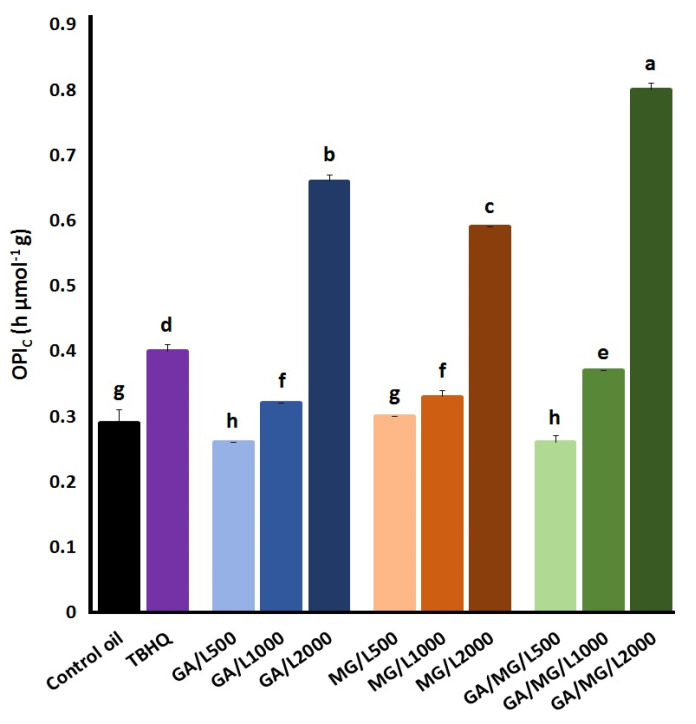
The overall carbonyls-based peroxidation indicator (OPI_C_) of the control sunflower/palm olein (65:35) oil containing TBHQ (1.2 mM) or GA and/or MG (1.2 mM) plus PC (500–2000 mg/kg) in terms of the ratio between the time threshold *t*_T_, resulting from the kinetic curves of LCD accumulation (Table 2), and [LCO] at the times. Means ± SD with the same lowercase letters are not significantly different at *p* < 0.05.

**Figure 5 foods-12-03560-f005:**
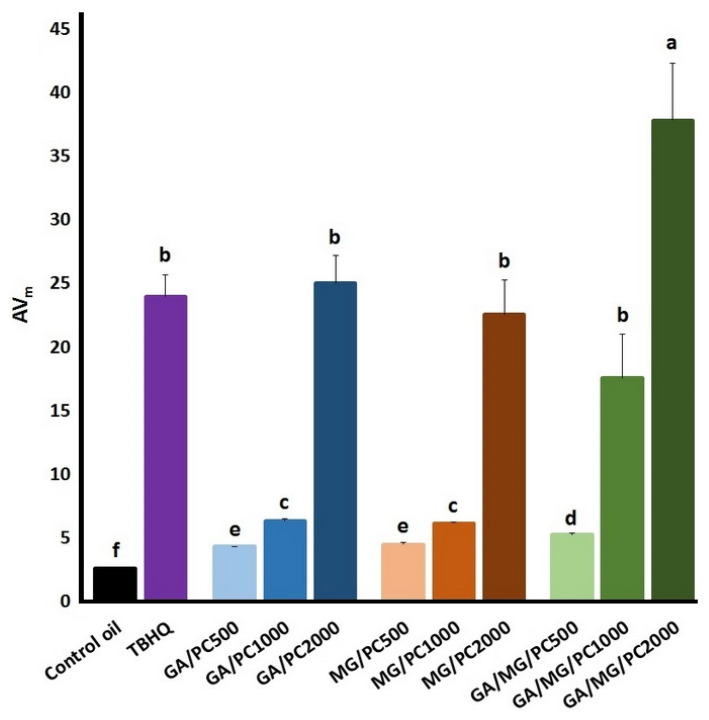
The quantitative measure of hydrolytic stability (AV_m_) from the accumulation curve of lipid-peroxidation organic acids during frying of the control sunflower/palm olein (65:35) oil containing TBHQ (1.2 mM) or GA and/or MG (1.2 mM) plus PC (500–2000 mg/kg) at 180 °C. Means ± SD with the same lowercase letters are not significantly different at *p* < 0.05.

**Table 1 foods-12-03560-t001:** Initial quality indicators of the oil samples.

	Oil Sample	
	Sunflower	Palm Olein
Major fatty acids (%)		
C14:0	–	1.0 ± 0.0
C16:0	7.0 ± 0.3	38.9 ± 0.2
C18:0	3.8 ± 0.1	4.4 ± 0.1
C18:1Δ^9^	25.5 ± 0.3	42.0 ± 0.6
C18:2Δ^9,12^	61.8 ± 0.4	12.2 ± 0.1
C18:3Δ^9,12,15^	0.2 ± 0.0	0.3 ± 0.0
Calculated oxidizability (COX) value	6.7 ± 0.1	1.7 ± 0.0
Peroxide value (PV, meq/kg)	1.53 ± 0.01	0.40 ± 0.00
Acid value (AV, mg/g)	0.15 ± 0.01	0.26 ± 0.03
Total tocopherols content (mg/kg)	490 ± 1	185 ± 1
Total phenolics content (mg/kg)	36.0 ± 1.0	53.1 ± 0.2

**Table 2 foods-12-03560-t002:** The kinetic parameters resulting from the accumulation curve of the total LCD content during frying of the control sunflower/palm olein (65:35) oil containing TBHQ or GA and/or MG plus PC at 180 °C ^1^.

Treatment	LCD_max_ (mM)	*t*_T_ (h)	*k*_c_ (/h)	*k*_d_ (/mM h)	*r*_n_ (/h)
Control oil	27.3 ± 0.5 ^c^	2.22 ± 0.06 ^g^	0.6639 ± 0.0081 ^ab^	0.0243 ± 0.0008 ^c^	0.1660 ± 0.0020 ^a^
TBHQ (1.2 mM)	26.1 ± 0.2 ^c^	4.48 ± 0.05 ^d^	0.3905 ± 0.0038 ^f^	0.0149 ± 0.0003 ^e^	0.0976 ± 0.0009 ^d^
GA (1.2 mM) + PC (mg/kg)					
500	17.4 ± 0.2 ^fg^	2.07 ± 0.02 ^h^	0.6790 ± 0.0101 ^a^	0.0391 ± 0.0009 ^a^	0.1697 ± 0.0025 ^a^
1000	18.4 ± 0.1 ^e^	3.07 ± 0.01 ^f^	0.4620 ± 0.0017 ^d^	0.0252 ± 0.0002 ^c^	0.1115 ± 0.0004 ^c^
2000	30.1 ± 0.5 ^b^	8.23 ± 0.12 ^b^	0.2415 ± 0.0005 ^h^	0.0080 ± 0.0002 ^g^	0.0604 ± 0.0001 ^f^
MG (1.2 mM) + PC (mg/kg)					
500	16.6 ± 0.1 ^h^	2.21 ± 0.01 ^g^	0.6521 ± 0.0018 ^b^	0.0393 ± 0.0002 ^a^	0.1630 ± 0.0017 ^a^
1000	17.2 ± 0.0 ^g^	3.02 ± 0.03 ^f^	0.4405 ± 0.0020 ^e^	0.0256 ± 0.0001 ^c^	0.1101 ± 0.0005 ^c^
2000	24.6 ± 0.0 ^d^	6.71 ± 0.01 ^c^	0.2421 ± 0.0001 ^h^	0.0099 ± 0.0000 ^f^	0.0605 ± 0.0000 ^f^
GA/MG 50:50 (1.2 mM) + PC (mg/kg)					
500	15.6 ± 0.4 ^i^	1.96 ± 0.05 ^h^	0.5075 ± 0.0008 ^c^	0.0326 ± 0.0008 ^b^	0.1269 ± 0.0002 ^b^
1000	17.8 ± 0.0 ^f^	3.63 ± 0.01 ^e^	0.3376 ± 0.0004 ^g^	0.0190 ± 0.0001 ^d^	0.0844 ± 0.0001 ^e^
2000	35.5 ± 0.6 ^a^	10.56 ± 0.11 ^a^	0.2067 ± 0.0003 ^i^	0.0058 ± 0.0001 ^h^	0.0517 ± 0.0001 ^g^

^1^ Means ± SD within a column with the same lowercase letters are not significantly different at *p* < 0.05.

**Table 3 foods-12-03560-t003:** The kinetic parameters resulting from the accumulation curve of the total LCO content during frying of the control sunflower/palm olein (65:35) oil containing TBHQ or GA and/or MG plus PC at 180 °C ^1^.

Treatment	LCO_max_ (μmol/g)	*t*_T_ (h)	*r*_max_ (μmol/g h)	*r*_n_ (/h)
Control oil	27.0 ± 0.8 ^b^	4.51 ± 0.08 ^c^	4.51 ± 0.18 ^a^	0.1668 ± 0.0019 ^a^
TBHQ (1.2 mM)	16.4 ± 0.0 ^g^	4.26 ± 0.04 ^d^	2.46 ± 0.03 ^c^	0.1500 ± 0.0016 ^b^
GA (1.2 mM) + PC (mg/kg)				
500	19.2 ± 0.1 ^d^	3.20 ± 0.05 ^h^	3.08 ± 0.05 ^b^	0.1607 ± 0.0037 ^a^
1000	18.3 ± 0.0 ^f^	3.45 ± 0.05 ^g^	2.23 ± 0.02 ^ef^	0.1215 ± 0.0012 ^c^
2000	13.2 ± 0.2 ^h^	4.80 ± 0.03 ^b^	1.39 ± 0.11 ^hi^	0.1049 ± 0.0014 ^d^
MG (1.2 mM) + PC (mg/kg)				
500	19.9 ± 0.1 ^c^	3.87 ± 0.05 ^e^	3.24 ± 0.05 ^b^	0.1631 ± 0.0016 ^a^
1000	18.1 ± 0.2 ^f^	3.59 ± 0.01 ^f^	2.21 ± 0.01 ^f^	0.1222 ± 0.0018 ^c^
2000	14.2 ± 0.6 ^h^	5.26 ± 0.37 ^b^	1.52 ± 0.09 ^h^	0.1073 ± 0.0020 ^d^
GA/MG 50:50 (1.2 mM) + PC (mg/kg)				
500	18.9 ± 0.1 ^e^	3.27 ± 0.01 ^h^	2.28 ± 0.02 ^de^	0.1204 ± 0.0008 ^c^
1000	19.9 ± 0.4 ^cd^	4.37 ± 0.15 ^cd^	1.96 ± 0.07 ^g^	0.0958 ± 0.0032 ^e^
2000	29.2 ± 0.3 ^a^	12.74 ± 0.16 ^a^	1.20 ± 0.05 ^i^	0.0411 ± 0.0013 ^f^

^1^ Means ± SD within a column with the same lowercase letters are not significantly different at *p* < 0.05.

## Data Availability

Data are contained within the article.

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
