# Peer review of "Frying Performance of Gallic Acid and/or Methyl Gallate Accompanied by Phosphatidylcholine"

_foods, 2023, doi:10.3390/foods12193560_

Round 1
Reviewer 1 Report
The manuscript entitled “Frying performance of gallic acid and/or methyl gallate accompanied by phosphatidylcholine” is a research article on the possibility of using gallic acid and its methyl ester in the accompanied phosphatidylcholine during the frying process. This work is a continuation of the author's work on the oxidation stability of oils with the addition of gallic acid and its ester during frying: Farhoosh, R.; Nyström, L. Antioxidant Potency of Gallic Acid, Methyl Gallate and Their Combinations in Sunflower Oil Triacylglycerols at High Temperature. Food Chemistry 2018, 244, 29–35. https://doi.org/10.1016/j.foodchem.2017.10.025.; Mansouri, H.; Farhoosh, R.; Rezaie, M. Interfacial Performance of Gallic Acid and Methyl Gallate Accompanied by Lecithin in Inhibiting Bulk Phase Oil Peroxidation. Food Chemistry 2020, 328, 127128. https://doi.org/10.1016/j.foodchem.2020.127128; Hosseinkhani, M.; Farhoosh, R. Kinetics of Chemical Deteriorations over the Frying Protected by Gallic Acid and Methyl Gallate. Sci Rep 2023, 13 (1), 11059. https://doi.org/10.1038/s41598-023-38385-2 and Hosseinkhani Abadchi, M., & Farhoosh, R. (2023). Investigation of the Impact of Gallic Acid, Methyl Gallate and their Combination on the Oxidative Stability of Frying Oli. Iranian Food Science and Technology Research Journal, 19(2), 383-397.https://doi.org/10.22067/ifstrj.2022.75114.1144. The author, similar to https://doi.org/10.1038/s41598-023-38385-2, described the influence of gallic acid and its ester with accompanied phosphatidylcholine for oil changes during frying. The manuscript possesses some weaknesses. The comments that follow are listed, not by importance, but by order of appearance:
1-Line 7 – two-time correspondence
2-Line 10 – abbreviation TBHQ not explained
3-Line 15-17 abbreviation rn, tT not explained
4-Line 91 – “Na2SO4” please add a subscript to the numbers
5- Line 118 – Please add the city of production to the bench-top fryer
6- In 2.5 Frying Test for LCO measurement lack of absorption wavelength
7 – Line 172-173 “The vegetable oils had the fatty acid compositions in agreement with those usually reported in literature.”
Could you add references? Additionally, it would be good to add that the parameters are in line with the Codex Alimentarius
8 – Line 178 – Please correct Cox to COX throughout the manuscript
9 – Line 193 – It should be methyl, not ethyl
10 – Figures – Please move the y-axis caption to the center and add inner y-axis markers.
11 – There is much information about the influence of gallic acid, methyl gallate, and TBHQ. Please add a comparison of your OSI results to other author
12 - Line 251 – The table title should be directly above the table. Move it to the next page
13 - Lines 293-303 -Probably too small spacing and font size. Please check it.
14 - Line 348 – “OPIC value” it should be OPIc
Discussion
1 - Almost all discussions are focused on the addition of phosphatidylcholine; please add information about the influence on gallic and methyl ester and extend their synergism effect with phosphatidylcholine. In addition, there is a lot of data on gallic acid and its esters during frying. Compare them to your research.
2- Line 340-341 “Apart from the negative effect on color, another concern on applying phospholipids in frying oils is their contribution to foaming, according to the Blumenthal’s surfactant theory of frying [43].”
What effect on the sensory properties of the tested oil (smell, color) does the addition of gallic acid, methyl gallate and phosphatidylcholine in the proposed concentrations have? Will it be acceptable to consumers?
3- Line 430-432 The present study was the first attempt to kinetically evaluate the protective effect of the two natural polyphenols gallic acid (GA) and methyl gallate (MG) accompanied by the major phospholipid phosphatidylcholine (PC) in vegetable oils for frying purposes”.
Both phenolic antioxidants are compounds found in plants. However, do you know if they are safe in your proposed concentrations?
References
References should be described according Foods requirements
Author 1, A.B.; Author 2, C.D. Title of the article. Abbreviated Journal Name Year, Volume, page range.
Please correct all references
Line 585-586 references should be separated from Disclaimer/Publisher’s Note
Author Response
1. Line 7 – two-time correspondence
It is not in the original manuscript and the Journal’s typesetter will correct it in the final version.
2. Line 10 – abbreviation TBHQ not explained
It was added.
3. Line 15-17 abbreviation rn, tT not explained
They were added.
4. Line 91 – “Na2SO4” please add a subscript to the numbers
It is correct in the original manuscript and the Journal’s typesetter will correct it in the final version.
5. Line 118 – Please add the city of production to the bench-top fryer
It was added.
6. In 2.5 Frying Test for LCO measurement lack of absorption wavelength
The absorbance of 420 nm was added.
7. Line 172-173 “The vegetable oils had the fatty acid compositions in agreement with those usually reported in literature.”
Could you add references? Additionally, it would be good to add that the parameters are in line with the Codex Alimentarius
The references were added.
8. Line 178 – Please correct Cox to COX throughout the manuscript
All were corrected.
9. Line 193 – It should be methyl, not ethyl
It was corrected.
10. Figures – Please move the y-axis caption to the center and add inner y-axis markers.
They were performed.
11. There is much information about the influence of gallic acid, methyl gallate, and TBHQ. Please add a comparison of your OSI results to other author
The comparisons are done in the discussion section through the references 48 – 50.
12. Line 251 – The table title should be directly above the table. Move it to the next page
It is correct in the original manuscript and the Journal’s typesetter will correct it in the final version.
13. Lines 293-303 -Probably too small spacing and font size. Please check it.
It is correct in the original manuscript and the Journal’s typesetter will correct it in the final version.
14. Line 348 – “OPIC value” it should be OPIc
It is correct in the original manuscript and the Journal’s typesetter will correct it in the final version.
Discussion
1. Almost all discussions are focused on the addition of phosphatidylcholine; please add information about the influence on gallic and methyl ester and extend their synergism effect with phosphatidylcholine. In addition, there is a lot of data on gallic acid and its esters during frying. Compare them to your research.
Kindly, as stated in the introduction section, our recent study has fully discussed on the frying performance of gallic acid and methyl gallate (Ref. 8: Hosseinkhani, M.; Farhoosh, R. Kinetics of chemical deteriorations over the frying protected by gallic acid and methyl gallate. Sci. Rep. 2023, 13, 11059.). Besides, everywhere needed in the present study, we have referred to the results of the previous research. So, the Discussion section has actually addressed to the frying performance of gallic acid and/or methyl gallate accompanied by phosphatidylcholine. Roughly two third of the Discussion section is regarding the interfacial phenomena to explain how the two antioxidants would have synergistically acted with the phospholipid.
2. Line 340-341 “Apart from the negative effect on color, another concern on applying phospholipids in frying oils is their contribution to foaming, according to the Blumenthal’s surfactant theory of frying [43].” What effect on the sensory properties of the tested oil (smell, color) does the addition of gallic acid, methyl gallate and phosphatidylcholine in the proposed concentrations have? Will it be acceptable to consumers?
As the respected reviewer has mentioned above, our research group has worked years on these two antioxidants in various conditions and vegetable oils. The concentration of 1.2 mM, which is about 200 ppm, for the antioxidants is quite low and according to the standard levels. So, there is no organoleptic problem with them. As for phosphatidylcholine, the concentrations lower than 2000 ppm have already been used in a number of studies referred in this study with no report on their undesirable sensory properties. Meanwhile, the potatoes fried in our study had no serious sensory problem, although we did carry out any sensory test. Undoubtedly, to commercialize such a treatment we must definitely do such studies in future.
3. Line 430-432 The present study was the first attempt to kinetically evaluate the protective effect of the two natural polyphenols gallic acid (GA) and methyl gallate (MG) accompanied by the major phospholipid phosphatidylcholine (PC) in vegetable oils for frying purposes”. Both phenolic antioxidants are compounds found in plants. However, do you know if they are safe in your proposed concentrations?
As explained above, the concentration of 1.2 mM, which is about 200 ppm, for the antioxidants is quite low and according to the standard levels.
References
References should be described according Foods requirements
Author 1, A.B.; Author 2, C.D. Title of the article. Abbreviated Journal Name Year, Volume, page range.
Please correct all references
They are correct in the original manuscript and the Journal’s typesetter will correct it in the final version.
Line 585-586 references should be separated from Disclaimer/Publisher’s Note
It is correct in the original manuscript and the Journal’s typesetter will correct it in the final version.
Reviewer 2 Report
The manuscript makes use of phosphatidylcholine along with gallic acid/methyl gallate to stabilize frying oils. The choice of oil is rather interesting as the oils used are not necessarily those that are most oxidizable. Furthermore, polar lipids are always measured when evaluating frying oil and this has not been done. Furthermore, gallic acid and methyl gallate are found naturally but what is used is not natural and in fact propyl gallate is another synthetic one along with those listed that has always been used.
Overall, the work has no real originality or novelty but one more confirmatory addition to what is know. Referencing has been seelctive and has avoided using prior references that have shown synergism although these might not have necessarily been for frying.
On the basis of the above, much improvement and also additional data on polar lipids are needed prior to possible further consideration of this submission.
Author Response
The manuscript makes use of phosphatidylcholine along with gallic acid/methyl gallate to stabilize frying oils. The choice of oil is rather interesting as the oils used are not necessarily those that are most oxidizable.
Polar lipids are always measured when evaluating frying oil and this has not been done.
We confirm that the total content of lipid-peroxidation polar compounds, namely TPC, is the most well-known analytical measure to evaluate the health of used frying oils. However, this measure has been shown to correlate very well with the simpler, quicker, and less expensive measure to determine, namely the total content of lipid-peroxidation conjugated dienes (LCD), which was employed in the present study.
Farhoosh, R. & Moosavi, S. M. R. Evaluating the performance of peroxide and conjugated diene values in monitoring quality of used frying oils. J. Agric. Sci. Technol. 11, 173–179 (2009).
Furthermore, gallic acid and methyl gallate are found naturally but what is used is not natural and in fact propyl gallate is another synthetic one along with those listed that has always been used.
Kindly, gallic acid and methyl gallate are natural compounds and the present study’s emphasis is on using natural antioxidants, which with no doubt are quite safer than synthetic ones like propyl gallate.
Overall, the work has no real originality or novelty but one more confirmatory addition to what is know.
As stated in the manuscript, the present study is the first attempt to kinetically evaluate the protective effect of the two natural polyphenols gallic acid and methyl gallate accompanied by the major phospholipid phosphatidylcholine in vegetable oils for frying purposes. The paper kinetically shows how interfacial phenomena may improve the frying performance of the antioxidants. Both the kinetic study used here and the interfacial phenomena, which have no long time with respect to bulk oils at normal storages, are quite novel under frying conditions, and literature review indicates no study of this kind.
Referencing has been selective and has avoided using prior references that have shown synergism although these might not have necessarily been for frying.
As stated the respected reviewer, synergism is a general issue and hundreds of papers may be found in literature on it. However, during a comprehensive literature review, only those quite relevant articles have been referred in this study.
Reviewer 3 Report
Is the abstract part complete ? I feel it is missing or incomplete.
What is the kinetic evaluation of the protective effect of gallic acid (GA) and methyl gallate (MG) in combination with phosphatidylcholine (PC) in vegetable oils for frying, and how does the concentration of PC influence frying performance, including the formation of reverse micelles and the antioxidant activity compared to the synthetic antioxidant TBHQ?
Secondly was there any optimization of combination of treatments taken into account while going for the comparison against the effective commercially proven antioxdiant ?
Author Response
Is the abstract part complete? I feel it is missing or incomplete.
The abstract was corrected.
What is the kinetic evaluation of the protective effect of gallic acid (GA) and methyl gallate (MG) in combination with phosphatidylcholine (PC) in vegetable oils for frying, and how does the concentration of PC influence frying performance, including the formation of reverse micelles and the antioxidant activity compared to the synthetic antioxidant TBHQ?
The kinetic evaluation provided a number of kinetic parameters and rate constants of the reactions occurred over the frying conditions accompanied by the antioxidants and the phospholipid, shown in Tables 2 and 3 as well as the figures. These data were adopted to interpret the probable events happened during the frying process.
Secondly was there any optimization of combination of treatments taken into account while going for the comparison against the effective commercially proven antioxdiant?
The present study demonstrated the synergism between gallic acid and/or methyl gallate and phosphatidylcholine that may potentially be used in the industry. Undoubtedly, to commercialize such a treatment we must definitely do some complementary studies in future.
Round 2
Reviewer 3 Report
After going through the MS i have the following queries. Please respond to these
1. What are the key advantages of deep-fat frying as a method of food preparation in the food industry?
2. Can you explain the economic significance of deep-fat frying in the USA and the rest of the world?
3. What are the potential flavor-related concerns associated with deep-fat frying?
4. How does the process of deep-fat frying affect the quality and stability of the frying oil?
5. What are some common approaches to stabilize frying oils, and why is the use of synthetic antioxidants like PG, BHA, BHT, and TBHQ being questioned?
6. What factors have led to a growing interest in replacing synthetic antioxidants with natural alternatives?
7. What are Gallic acid (GA) and methyl gallate (MG), and what is their relevance as natural phenolic antioxidants?
8. What is the significance of studying the performance of GA and MG along with PC in an institutional deep-fat frying condition?
9. How will the kinetics of chemical deteriorations be measured in the study, and what is the comparison being made with the synthetic antioxidant TBHQ?
10. What were the sources of the vegetable oils used in this study, and were any antioxidants added to these oils?
11. How were the oil samples stored prior to analysis, and at what temperature?
12. What chemicals and solvents were used in the study, and where were they purchased from?
13. Can you explain the process of preparing fatty acid methyl esters (FAME) and how they were analyzed?
14. What methods were used to determine the oxidizability (Cox) value, peroxide value (PV), acid value (AV), total tocopherols, and total phenolics in the oil samples?
15. What instrument was used to measure the oxidative stability index (OSI) of the oil treatments, and what conditions were used for this measurement?
16. Describe the frying procedure used in the study, including details about the type of potatoes, oil treatments, and frying equipment.
17. What specific tests were conducted to monitor the total contents of lipid-peroxidation conjugated dienes (LCD) and carbonyls (LCO) in the oil treatments during frying?
18. What parameter was used to quantify the hydrolytic stability of the frying system based on the accumulation of organic acids (mg/g) over time?
19. How was the statistical analysis conducted for the data obtained in this study, and what significance level was used to determine statistically significant differences?
20. How did the antioxidative treatments affect the level of LCDmax in the control oil, and what does this change signify?
21. How does the major phospholipid PC in vegetable oils differ from other raised phospholipids like PE and PS in terms of its reactivity with Maillard reaction substrates?
22. What is the significance of PC's quaternary amine choline in relation to the Maillard reaction and its suitability for frying purposes?
23. How does the presence of phospholipids with primary amine groups, such as PE and PS, affect the color of frying oils, and what role do they play in lipid oxidation?
24. Despite not serving as Millard reaction substrates, why might phospholipids with primary amine groups still be valuable in inhibiting lipid oxidation?
25. What concerns are associated with using phospholipids in frying oils, particularly regarding foaming and Blumenthal’s surfactant theory of frying?
26. Can you explain why phospholipids may accelerate lipid oxidation by reducing surface tension in the oil system?
27. How did the addition of 500 mg/kg PC alone affect the oxidative stability index (OSI) values in the Rancimat test, and what might explain this result?
28. What role do the negative charges on the phosphate headgroup of phospholipids play in their prooxidative/antioxidative activity, especially concerning metal-binding capacity?
29. How does the formation of reverse micelles by phospholipids in bulk oils affect the rate of lipid oxidation, and what is the critical micelle concentration (CMC) of PC?
Author Response
- What are the key advantages of deep-fat frying as a method of food preparation in the food industry?
It has been explained in the first paragraph of the section “Introduction” (Ref. 1).
- Can you explain the economic significance of deep-fat frying in the USA and the rest of the world?
It has been explained in the first paragraph of the section “Introduction” (Ref. 1).
- What are the potential flavor-related concerns associated with deep-fat frying?
It has been explained in the first paragraph of the section “Introduction” (Ref. 2).
- How does the process of deep-fat frying affect the quality and stability of the frying oil?
It has been explained in the first paragraph of the section “Introduction” (Ref. 2).
- What are some common approaches to stabilize frying oils, and why is the use of synthetic antioxidants like PG, BHA, BHT, and TBHQ being questioned?
The first sentence of the second paragraph of the section “Introduction” was corrected.
- What factors have led to a growing interest in replacing synthetic antioxidants with natural alternatives?
It has been explained in the second paragraph of the section “Introduction” (toxicity and carcinogenicity, Ref. 3).
- What are Gallic acid (GA) and methyl gallate (MG), and what is their relevance as natural phenolic antioxidants?
Please refer to the first sentence of the third paragraph of the section “Introduction” (Ref. 5). Meanwhile, some little explanation on their chemical structure was provided.
- What is the significance of studying the performance of GA and MG along with PC in an institutional deep-fat frying condition?
As explained in the 4th paragraph of the section “Introduction” (Ref. 15), only a small-scale preliminary frying test had been applied in this respect. The present study involves an actual frying process.
- How will the kinetics of chemical deteriorations be measured in the study, and what is the comparison being made with the synthetic antioxidant TBHQ?
Please refer to the section “2.6. Kinetic data analyses” (Ref. 24-26). The LCD- and LCO-based kinetic evaluations actually provided a number of kinetic parameters and rate constants of the reactions occurred over the frying conditions accompanied by the antioxidants (GA, MG, and TBHQ) and the phospholipid, shown in Tables 2 and 3 as well as the figures. These data were adopted to interpret the probable events happened during the frying process.
- What were the sources of the vegetable oils used in this study, and were any antioxidants added to these oils?
All have been explained in the section “2.1. Materials”.
- How were the oil samples stored prior to analysis, and at what temperature?
Please refer to the section “2.1. Materials”.
- What chemicals and solvents were used in the study, and where were they purchased from?
Please refer to the section “2.1. Materials”.
- Can you explain the process of preparing fatty acid methyl esters (FAME) and how they were analyzed?
Please refer to the section “2.2. Initial quality indicators of the vegetable oils”.
- What methods were used to determine the oxidizability (Cox) value, peroxide value (PV), acid value (AV), total tocopherols, and total phenolics in the oil samples?
Please refer to the section “2.2. Initial quality indicators of the vegetable oils” (Refs. 17-21).
- What instrument was used to measure the oxidative stability index (OSI) of the oil treatments, and what conditions were used for this measurement?
Please refer to the section “2.3. Oxidative stability index (OSI)”.
- Describe the frying procedure used in the study, including details about the type of potatoes, oil treatments, and frying equipment.
Please refer to the section “2.4. Frying procedure”.
- What specific tests were conducted to monitor the total contents of lipid-peroxidation conjugated dienes (LCD) and carbonyls (LCO) in the oil treatments during frying?
Please refer to the section “2.5. Frying tests” (Refs. 19, 22, and 23).
- What parameter was used to quantify the hydrolytic stability of the frying system based on the accumulation of organic acids (mg/g) over time?
Please refer to the section “2.5. Frying tests” (Ref. 19).
- How was the statistical analysis conducted for the data obtained in this study, and what significance level was used to determine statistically significant differences?
Please refer to the section “2.7. Statistical analysis”.
- How did the antioxidative treatments affect the level of LCDmax in the control oil, and what does this change signify?
As clearly explained in the section “3.3. Kinetics of change in the total LCD content”, any antioxidant treatments affect the two rate constants kc and kd, and a balance between them basically determine the level of LCDmax, indicating how an antioxidant can change the rates of formation and decomposition of lipid hydroperoxides.
- How does the major phospholipid PC in vegetable oils differ from other raised phospholipids like PE and PS in terms of its reactivity with Maillard reaction substrates?
As explained in the first paragraph of the section “4. Discussion” (Refs. 15 and 42), PC has no primary amine group, like PE and PS, which is necessary to take part in the Maillard reaction.
- What is the significance of PC's quaternary amine choline in relation to the Maillard reaction and its suitability for frying purposes?
The quaternary amine choline which is a moiety of the chemical structure of PC is not able to take part in the Maillard reaction, and therefore, make browning.
- How does the presence of phospholipids with primary amine groups, such as PE and PS, affect the color of frying oils, and what role do they play in lipid oxidation?
PE and PS participate in the Maillard reaction through their primary amine groups and discolor frying oils and the foods are being fried.
- Despite not serving as Millard reaction substrates, why might phospholipids with primary amine groups still be valuable in inhibiting lipid oxidation?
According to the mechanisms mentioned in the 4th paragraph of the section “Introduction” (Refs. 9-13), phospholipids are able to inhibit lipid peroxidation. Interestingly, the Maillard reaction provides some reductants as potent antioxidants as well. However, at frying conditions, which is so harsh, its browning aspect negatively affect the application of PE and PC for such a purpose.
- What concerns are associated with using phospholipids in frying oils, particularly regarding foaming and Blumenthal’s surfactant theory of frying?
Foaming over the frying process provides thin layers of the oil that are burnt and oxidized more easily. This makes the system more unstable oxidatively.
- Can you explain why phospholipids may accelerate lipid oxidation by reducing surface tension in the oil system?
Decrease in surface tension facilitates the incorporation of oxygen into the oil. The increased concentration of oxygen in the oil naturally enhances oxidation reactions.
- How did the addition of 500 mg/kg PC alone affect the oxidative stability index (OSI) values in the Rancimat test, and what might explain this result?
As explained above and also in the second paragraph of the section “4. Discussion”, this concentration must have been caused to incorporate more oxygen into the system.
- What role do the negative charges on the phosphate headgroup of phospholipids play in their prooxidative/antioxidative activity, especially concerning metal-binding capacity?
As fully explained in the third paragraph of the section “4. Discussion”, phospholipids might not chelate metals strongly, which this will lead to their more solubility. Clearly, such metals act as prooxidant in any lipid system.
- How does the formation of reverse micelles by phospholipids in bulk oils affect the rate of lipid oxidation, and what is the critical micelle concentration (CMC) of PC?
Kindly, this issue has been explained comprehensively in the section “4. Discussion”.